# The Quality and Health-Promoting Value of Meat from Pigs of the Native Breed as the Effect of Extensive Feeding with Acorns

**DOI:** 10.3390/ani11030789

**Published:** 2021-03-12

**Authors:** Magdalena Szyndler-Nędza, Małgorzata Świątkiewicz, Łukasz Migdał, Władysław Migdał

**Affiliations:** 1Department of Pig Breeding, National Research Institute of Animal Production, Krakowska 1, 32-083 Balice, Poland; magdalena.szyndler@izoo.krakow.pl; 2Department of Animal Nutrition and Feed Science, National Research Institute of Animal Production, Krakowska 1, 32-083 Balice, Poland; malgorzata.swiatkiewicz@izoo.krakow.pl; 3Department of Genetics, Animal Breeding and Ethology, University of Agriculture in Krakow, al. 29 Listopada 46, 31-425 Kraków, Poland; 4Department of Animal Product Technology, Faculty of Food Technology, University of Agriculture in Kraków, ul. Balicka 122, 31-149 Kraków, Poland; wladyslaw.migdal@urk.edu.pl

**Keywords:** Złotnicka Spotted pig breed, fattening system, acorns, meat quality, lipid quality indices

## Abstract

**Simple Summary:**

Modern society is paying more attention to the quality of meat and its production system. Therefore, in this study, we analyzed the possibility of using a native pig breed—Złotnicka Spotted. This breed’s popularity has increased in Poland as it may be a good source of valuable meat. Compared to the meat frompigs of the same breed following intensive fattening, the meat from free-range pigs extensively fed on silage and small amounts of acorns is characterized by a higher content of fat, which acts as a carrier for flavor and, indirectly, juiciness. Moreover, rearing systems with traditional feeding, e.g., with the use of acorns, may have a positive effect on meat quality, dietetic value, and, therefore, consumer acceptance of the product.

**Abstract:**

The popularity of meat from animals of native breeds is growing all over the world, due to consumer belief regarding its higher quality compared to meat from industrial farm animals. In addition, the living conditions (welfare) are of great importance for consumers. We observed the effect of different ways of keeping and feeding pigs of the same conservative breed on the quality of meat and its health benefits. The aim of the study was to compare the meat quality from pigs of the native Złotnicka Spotted breed, fattened intensively or extensively (with conventional farm-produced compound feed and acorns). The meat from free-range pigs extensively fed on silage and small amounts of acorns was characterized by a higher content of fat, which acts as a carrier for flavor and juiciness, as well as higher monounsaturated fatty acid (MUFA) content (*p* ≤ 0.05) and lower atherogenic, thrombogenic, and peroxidability indices (*p* ≤ 0.05). It may be stated that the meat quality of the native pig breed is significantly dependent on the housing and feeding method. A more beneficial effect on the quality of meat and its dietetic value, as well as its susceptibility to rancidity, can be obtained throughextensive pig feeding with roughage and the addition of acorns.

## 1. Introduction

In the last few years, there has been growing interest in animal products originating from animal systems that could be considered “natural” or “traditional”. Some consumers consider the traditional extensive pig production system as synonymous with product quality [1]. Intensive animal production systems, such as those practiced in pig and poultry production, are being questioned because of impaired welfare, as these animals are kept in total confinement, representing an unnatural environment. More and more consumers are revealingan underlying desire to obtain animal products with certainsocial attributes or assurances, such as protection of the environment, ethical rearing of the animals, and development of the small farmer. An increasing numberof consumers arewilling to pay more for pork products with social assurances, in addition to associating the free-range products with a better taste and nutritional value than conventionally produced products [2]. This obviously opens niche markets for naturally produced pork. The traditional fattening of native breed pigs in Poland is predominantly different from the fattening of lean pigs; it is less intensive, and animals have more freedom to express natural behavior patterns. In the traditional system, pigs are fattened to reach higher weight gains and higher fat content because the objective is not the quantity of meat but the quality and maturity of meat and fat. Another frequent component of traditional fattening is the high variation of feed ingredients, for example, by the dietary inclusion of seasonal roughages (forages, silages, or root crops). One of the best-known breeds raised in the extensive system is theIberianpig. They are fed with acorns, which is typical for the traditional system of Iberian pig farming, using the pasturelands of oaks in the southwestern part of the Iberian Peninsula, where, during the *montanera* season, the pigs are fed exclusively on fallen acorns and grass. In contrast, Złotnicka Spotted pigs have practically no access to range freely and are not fed on acorns. However,knowing the clear improvement in the quality and palatability of the meat fromIberian native pigs fed on acorns [3], we decided to add acorns to the diet of Polish native pigs. The oak species native to Poland are pedunculate oak (*Quercus robur L*.) and sessile oak (*Quercus petra (Matt.)*). *Q. robur* is found all over Poland up to analtitude of 600 m. The distribution of *Q. petraea* is similar, except that it is not found in the northeastern region of lowland Poland, at an altitude up to 750 m [4]. Another common exotic oak taxon is the introduced northern red oak (*Q. rubra L.*), which is planted in forests and parks and often naturalized [5]. Under Polish conditions, oaks bear fruit every 5–8 years. Fruits mature in September–October, after which they fall. The weight of 1 hl of acorns exceeds 80 kg (http://www.encyklopedia.lasypolskie.pl/ accessed on 12 March 2021). The acorns gathered in western Europe are rich in α-tocopherol (10–15 mg/kg dry matter (DM)), γ-tocopherol (60–84 mg/kg DM), and tannins (0.8–1.1 g/kg DM), with the total phenolic compounds amounting to about 9.8–14 g/kg DM [6,7,8]. In Poland, four tocopherols (Ts) (α-T, β-T, γ-T, and δ-T) in the oak fruits of *Q. rubra* and *Q. robur* were identified. The *Q. robur* acorns had abundant γ-T (28.18 ± 6.7 mg/100 g DM), while, in *Q. rubra*, β-T was predominant (17.28 ± 2.91 mg/100 g DM) [9]. Moreover, depending on variety and occurrence area, these acorns contain from 2.97% to 3.48% DM tannins and from 4.29% to 4.58% DM total phenolics [5]. The tannin content of acorns from Polish oaks can reach 7%, but far greater amounts are found in the shell—42.2 g/kg DM [10,11]. Tannins have a high antioxidant capacity; however, binding to proteins reduces their digestibility [12,13]. Both tocopherols and phenolic compounds show great ability to stimulate the antioxidant and free-radical-scavenging activities, with potential benefits for human health, particularly contributing to the prevention of degenerative processes [14]. The acorn is an energy food rich in fat and carbohydrates; however, it has very low protein content and its amino-acid profile indicates that lysine is the main limiting amino acid [15]. In addition, acorns have a high concentration of monounsaturated fatty acids, particularly oleic acid C18:1 n-9 (60–65%), followed by palmitic acid C16:0 (about 13–14%) and linoleic acid C18:2 n-6 (15–18%) [13,16]. The fatty acid content of acorns influences the quality of meat and meat products from the pigs fed in this manner during the last phase of fattening [17]. In addition to modifying the level of meat fatty acids, acorn feeding has an effect on the health of pigs. A study by Salajpal et al. [18] suggested that acorns fed during the last phase of fattening reduce gastrointestinal nematodes and, thus, lower the need for deworming the pigs.

According to the available literature, we hypothesized that the addition of acorn to feed of Polish native pigs may have a beneficial influence on meat quality and fatty acid composition.

The aim of the study was to compare the meat quality of native Złotnicka Spotted pigs raised in two different systems: intensive fattening with a complete diet or extensive feeding with conventional farm-produced compound feed and acorns, as well as access to forage and to range freely.

## 2. Materials and Methods

### 2.1. Animals, Feeding, and Housing System

The study involved 13 fattening gilts of the Złotnicka Spotted breed, which were reared on two farms in different fattening systems. All animals originated from farms included in the genetic resource conservation program for Złotnicka Spotted pigs.

The first farm, belonging to the Experimental Station of the National Research Institute of Animal Production in Chorzelów, kept six gilts from a herd in the Wielkopolska province. Animals were fattened for 9 months under intensive farm conditions (control group), without outdoor access, until attaining 100 kg of body weight. Animals were fed ad libitum with a complete diet formulated to meet the nutrient requirements for grower (from 30 to 80 kg of body weight) and finisher pigs (from 80 to 100 kg of body weight). The feed mixtureconsisted of wheat, maize, extracted soybean meal, rapeseed cake, dehulled sunflower seeds, wheat bran, and mineral/vitamin supplements. The diet contained 170–162 g/kg crude protein, 13.5–13megajoules of metabolizable energy (MJ ME), 1.1–0.93% Lys, 46–56 g/kg fat, 50–48 g/kg crude fiber, and 59.6 g/kg nitrogen-free extract (NFE) in the grower and finisher phases.

In the second farm under conservation, located in the Kujawsko-Pomorskie province, seven gilts were kept under extensive free-range conditions (experimental group) and fattened until 12 months of age. Animals were fed a compound feed at 2 kg/head/day and had access to roughage in the form of grass and whole-crop maize silage, as well as hay. The produced feed mixture consisted of barley (23%), wheat (23%), oats (23%), maize (10%), and a mixture of expeller black cumin, mustard, linseed, and rapeseed, which constituted 21%. This diet contained 150 g/kg crude protein, 44 g/kg fat, 13 MJ ME, 51 g/kg crude fiber, and 639.8 g/kg NFE. During the last 2 months of fattening, acorns (whole fruits) were added to the diet at 5%. Acorns were harvested from pedunculate (*Quercus robur L*.) and sessile oaks (*Quercus petraea (Matt.)*) growing in the Kujawsko-Pomorskie province. Acorns contained 68.50% dry matter, 3.62% protein, 3.45% fat, 1.22% ash, 1.82% fiber, and 54.99% NFE.

Differences in body weight were due to most pigs under the preservation program being kept in extensive conditions. For the purpose of this research, we kept pigs under intensive conditions for comparison with the extensive feeding regime and with other intensively kept pig breeds. Because animals were kept on commercial farms, ethical approval was unnecessary, as we only collected samples from carcasses in the slaughterhouse.

Upon reaching their designated age or body weight (depending on the type of fattening), all animals were slaughtered in a specialized slaughterhouse. After slaughter and 24 h chilling at 4 °C, a 100 g slice of meat was cut from the middle part of the *m. longissimus dorsi* of each carcass (between the last thoracic vertebra and first lumbar vertebra), vacuum-packed into a plastic bag, labeled, and frozen at −18 °C until analyses. Carcass weight, age of animals, dressing percentage, and daily weight gain were determined during the experiment.

#### 2.1.1. Chemical Analyses and Calculations

The samples of the finishing diets were analyzed for the content of basic chemical components, the content of amino acids per 100 g protein, and the fatty acid profile. The samples of meat taken from the middle part of the *m. longissimus dorsi* were analyzed for basic chemical composition, fatty acid profile, and color.The basic chemical compositions of the feed mixtures were analyzed according to the methods of the Association of Official Analytical Chemists (AOAC) [19].

Amino acids were determined using the HPLC method after acid hydrolysis of samples in 6N hydrochloric acid at 110 °C for 22 h in acolor reaction with the ninhydrin reagent, using an AAA 400 automatic analyzer (Ingos Ltd., Prague, Czech Republic). Sulfur amino acids were estimated after initial peroxidation with performic acid to cysteic acid and methionine sulfone. Tryptophan content was estimated after alkaline hydrolysis of samples in BaOH solution and precipitation of bar ions using sulfuric acid [20].

The fatty acid profile was determined after 2 weeks of freezing at −20 °C, immediately after thawing, using a CP-Wax 58 capillary column (Varian BV, Middelburg, The Netherlands; 25 m, 0.53 mm, DF = 1 µ, carrier gas = helium, 6 mL/min), with a column oven temperature program from 90 °C to 200 °C, by using a Varian 3400 gas chromatograph (Varian Associates Inc., Walnut Creek, CA, USA) equipped with a Varian 8200 CX autosampler (200 °C), an FID (Flame Ionization Detector) detector (260 °C), and Star Chromatography Workstation software. All analyses were performed in duplicate, and the mean values are given.

The color of the fresh meat was estimated after 24 chilling using the CIElab color space (L*, a*, b*) (Commission internationale de l’éclairage, CIE) system using a chroma meter CR-410 (Minolta Co; Ltd., Suita, Osaka,Japan).

The iodine value (IV) of fat was calculated using the following equation [21]:IV = (g/100 g) = (C16:1) × 0.95 + (C18:1) × 0.86 + (C18:2) × 1.732 + (C18:3) × 2.616 + (C20:1) × 0.785 + (C22:1) × 0.723.

Lipid quality indices, i.e., the atherogenicity index (AI) and thrombogenicity index (TI), were calculated according to the Ulbricht and Southgate [22] equations, where MUFA denotes monounsaturated fatty acids, and PUFA denotes polyunsaturated fatty acids.
AI = (C12:0 + (4 × C14:0) + C16:0)/(n − 6 PUFA + n− 3 PUFA + MUFA),
TI = (C14:0 + C16:0 + C18:0)/((0.5 × MUFA) + (0.5×n – 6 PUFA) + (3 × n − 3 PUFA) + n − 3/n − 6 PUFA).

The hypocholesterolemic/hypercholesterolemic ratio (h/H) was calculated according to theFernández et al. [23] formula.
h/H = (C18:1 + C18:2 + C18:3 + C20:3 + C20:4 + C20:5 + C22:4 + C22:5 + C22:6)/(C14:0 + C16:0).

The peroxidability index (PI) was calculated (Σg/100 g) according to Arakawa and Sagai [24]:PI = (% monoenoic × 0.025) + (% dienoic × 1) + (% trienoic × 2) + (% tetraenoic × 4) + (% pentaenoic × 6) + (% hexaenoic × 8).

#### 2.1.2. Statistical Analysis

The results were statistically analyzed using the Statistica package, version 12. The effect of the type of fattening (intensive or extensive) on the production parameters of the pigs and the chemical composition, fatty acid content, and health benefits of their meat were determined. The animal carcass weight was grouped into threelevels: (1) >90 kg, (2) 90.1–96 kg, and (3) >96 kg.The results were analyzed using two-way ANOVA according to the following model:Y_ij_ = µ + a_i_ + b_j_ + e_ij_,
where Y_ij_ represents the observations, µ is the overall mean, a_i_ is the fixed effect of type of fattening (i = 2), b_j_ is the fixed effect of carcass weight (j = 3), and e_ij_ represents random error. Significant differences between the groups were considered at *p* ≤ 0.05 using the LSD (Least Significant Difference) test.

## 3. Results

The complete diet (control feed mixture) used in conventional intensive fattening differed in terms of protein and fat content from the farm-produced compound feed with acorns (experimental feed mixture). The extensive fattening diet was lower in both protein (by 8%) and fat (by 20%). The diets did not differ in the total amount of unsaturated fatty acids (UFAs), but showed distinct differences in the proportions of MUFAs and PUFAs (Table 1).

The experimental diet was characterized by a lower MUFA content (32 vs. 56 g/100 g) and the difference was mainly due to the lower amount of oleic acid (C18:1). Moreover, this feed had a higher PUFA content (52 vs. 32 g/100 g), which resulted from greater amounts of linoleic acid (C18:2) and linolenic acid (C18:3) (by 52% and >300%, respectively). The n-6/n-3 PUFA ratio in the experimental diet was over 52% lower than in the control diet. The amount of saturated acids was 32.5% greater in the experimental diet.

Table 2 presents the amino-acid content per 100 g of protein in the intensive and extensive fattening diets. Protein from the compound feed fed in the extensive system had a slightly higher content of lysine (3.49 and 2.74, respectively) and methionine (1.10 and 0.71, respectively)—the exogenous amino acids essential for growing pigs—and a slightly higher amount of valine. Protein from the extensive feeding diet was lower in serine, glutamine, and histidine, but these differences were small and did not exceed 1%. With such small differences, it can be concluded that the protein quality in the two feed mixtures was comparable.

The control pigs were reared until 100 kg of body weight, which occurred with a daily gain of 580 g when the pigs were over 9 months old (Table 3). The extensively fed pigs were fattened to about 120 kg of body weight, which they reached at 12 months of age with an average daily gain of 340 g. Dressing percentage was significantly higher for the extensively compared to the intensively fattened pigs (79% vs. 76%).

The meat quality of the Złotnicka Spotted pigs reared according to a different fattening procedures did not differ significantly in the content of dry matter, protein, and ash, which also included minerals (Table 4). The meat fromextensively fattened pigs had a 66% higher fat content (*p* ≤ 0.05), which is beneficial in terms of product taste, juiciness, and tenderness. Fattening method had no significant effect on the meat color.

The way the Złotnicka Spotted fatteners were fed and raised had a significant effect on the composition of the meat fatty acids (Table 5). The meat from extensively fed pigs contained a greatertotal FAs (*p* ≤ 0.05), including MUFAs (*p* ≤ 0.05), particularly C18:1 n-7 (*p* ≤ 0.05) and C16:1 n-9 (*p* ≤ 0.05), but alower total PUFAs, although a statistically significant difference was observed only for n-3 PUFAs (*p* ≤ 0.05). Compared to the meat fromintensively fattened pigs of the same breed, the meat from extensively fed pigs contained fewerlong-chain unsaturated fatty acids and exhibited a slightly higher n-6/n-3 PUFA ratio (*p* > 0.05); however, from the statistical and nutritional view point, the difference between the n-6/n-3 ratios of 10 and 13 was of noreal significance.

The meat fat from the Złotnicka Spotted pigs fed extensively with supplemental acorns was characterized by a significantly lower (*p* ≤ 0.05) thrombogenicity index (TI). With regard to the effect of fatty acid content on human health, i.e., related to myristic C14:0, palmitic C16:0, and stearic C18:0 acids, the meat from extensively fed pigs contained significantly less C14:0 (*p* ≤ 0.05) (Table 5).

The peroxidability index (PI) was 25.8% lower (*p* ≤ 0.05) in the meat from extensively fed pigs (Table 5). The iodine value (IV) of fat was comparable in both groups, regardless of the fattening method.

## 4. Discussion

There are three native pig breeds—Puławska, Złotnicka White, and Złotnicka Spotted—in Poland which are valuable due to the distinguishing quality traits of their meat. The meat obtained from these animals is characterized by a high intramuscular fat (IMF) content and a desirable color, texture, and juiciness,giving rise to a higher culinary value than mass-produced pork [25]. However, Złotnicka Spotted pigs, due to their slower growth rate, are less popular, but still worthy of interest. Our comparative test confirmed the beneficial effect of traditional extensive fattening on the quality of meat fromthese pigs, although their weight gains were significantly lower when compared to pigs fattened intensively indoors.

### 4.1. Fattening System and Fattening Performance

The Mediterranean pig breeds Alentejano and Iberian, raised in the southwest of the Iberian Peninsula, are characterized by a lower rate of growth compared to commercial breeds such as Duroc and Landrace, as well as a higher content of subcutaneous fat and high-quality meat resulting from the higher content of intramuscular fat. These pigs are allowed to roam freely over large areas, where they forage on grass and acorns in the *montanera* fattening system. The meat fromthese breeds is valued for its high quality and suitability for the production of local products [26,27]. In this traditional system for producing Iberian pigs, the diet of pigs in the finishing phase (up to 160 kg body weight), which lasts 3 to 5 months (from November to March), is based on ad libitum feeding with acorns and grass [28]. Margeta et al. [8] used a similar feeding system for Black Slavonian (Crna slavonska) pigs and fattened them until 130–150 kg of body weight for at least 18 months. The pigs were maintained on pasture where they consumed forage, whereasthey were fed acornsin the winter, which the authors considered one of the most important elements in the extensive system. The specific chemical composition and antioxidant properties of acorns contribute to the outcome of fattening. The most popular species of oak in Croatia, similar to Poland, is *Quercus robur*, which yields about 270 kg/ha of acorns [29]. Acorns are fed from December to February. Because oak acorns contain more than 10% fat [15], acorns fed to pigs at 5 kg/day increased the average daily weight gains, resulting in a considerably higher (*p* < 0.01) final body weight of the pigs [8]. Similar results were obtained by García-Valverde et al. [15] and Rodríguez-Estévez et al. [30]. Tejerina et al. [31] pointed out that results are not repeatable due to the variable nature of extensive pig production (effect of season and year), which is reflected in the variation in diet composition (acorns and grass). In our study, adding acorns at 5% (per 2 kg of compound feed/head/day) proved inadequate to increase the fat and energy content of the feed and, thus, to improve daily weight gains in the extensively fed pigs. The daily gains obtained in this group of animals were slightly lower than the breed average of 433 g/day [32]. Furthermore, Nieto et al. [33] reported a negative effect of acorn shells on the protein digestibility and average intestinal digestibility of lysine (0.601), the first limiting amino acid for pigs. In turn, Kilic et al. [10] concluded that alevel of dietary tannin above 2–3% may decrease N absorption in the duodenum, feed intake, and digestibility. In view of the above, the observed differences in daily gain between free-range and indoor-reared pigs wereprobably due to the lower feeding intensity (2 kg/head/day of feed mixture), the inclusion of whole acorns with shells, and the increased intake of nutrients and energy to meet the requirements of free-range management, i.e., physical activity [34] and the need to adapt to lower temperatures and weather conditions [35]. It should be mentioned that the acorns from Polish pedunculate and sessile oaks, fed to Złotnicka Spotted pigs, had a lower content of protein (3.62%) and fat (3.45%) compared to acorns from Mediterranean oaks, among which the cork oak and the holm oak are predominant. The fat from the experimental acorns was characterized by a high content of C18:1 n-9 fatty acid (59.36%, Table 1). Özcan [36] found 7.41% fat (dry wt.%) and 48.25% oleic acid in acorns from northern red oak (*Quercusrubra* L.) growing in Turkey. Akcan et al. [37], who analyzed the fatty acid profile of different oak species from the Caceres region (Spain), showed the level of C18:1 n-9 acid to range from 48.02% (kermes oak, *Quercuscoccifera L.)* to 65.83% (holm oak, *Quercus ilex* L. subsp. *ballota* [Desf.]). The same authors also observed that the ether extract ofoak acorns from the northern part of the Cacares region had a lower C18:1 n-9 acid content and a higher C18:2 (n-6) acid content compared to oak acorns from the southern part of the region. Özcan et al. [38] also reported the effect of oak species on the protein content and amino-acid profile. According to the same authors, the protein content of the acorns ranged from 2.75% to 8.44% and the content of amino acids varied greatly according to oak species. The above differences in the chemical composition of oak acorns, resulting from species and climatic differences, may explain the varying results obtained by different authors whenfeeding oak acorns to animals.

### 4.2. Fattening System vs. Meat Color and Fat Content

A higher activity of pigs reared in the outdoor system probably stimulates muscle fiber development [39], as well as a higher myoglobin content in meat [40]; consequently, the muscles of outdoor reared pigs show more type I fibers than thoseof pigs bred and housed indoors [41]. The analysis of meat color in the present experiment did not show any significant differences in the intensity of redness between intensively and extensively fattened pigs, which is in contradiction to the values often available in the literature reporting higher meat redness in animals managed with access to outdoor yard and green forage [2,42]. The meat color parameters observed in the present study are consistent with the findings of Tejeda et al. [43], who found no statistically significant differences in the lightness, redness, and yellowness of *m. longissimus dorsi* between free-range and indoor-raised pigs. Regardless of the fattening method, the lightness value obtained for the meat from the Złotnicka Spotted pigs was typical of the breed and consistent with the results of other authors [44]. The meat color of these pigs showed considerably greater redness and yellowness compared to the values of these parameters observed in the breeds of lean pigs raised in Poland [45], which is probably due to the greater amount of fat in meat and the greater number of type I red muscle fibers characteristic ofnative Polish pig breeds [46].

The extensive fattening of the Złotnicka Spotted pigs, which had access to grass silage and were supplemented with compound feed and acorns in a free-range environment until 120 kg of body weight, contributed to a significant increase in the meat fat content,which was also a function of the animals’ age. As reported in Table 1, for experimental feeding, the level of PUFAswashigher compared to control. This may have caused significant differences in the fatty acid profile of the meat and a possible decrease in carcass quality. However, the results of our study indicate that the content of PUFAs in the meat from extensively fed pigs was not significantly different compared to the control group (Table 5). Likewise, Iberian pigs raised in the *montanera* system showed a higher meat fat content [47]. As is known, pigs receiving inadequate dietary protein in relation to energy deposit excess energy as body fat [48]. Doran et al. [49] suggested that, with low-protein diets, lipogenic enzymes are expressed more readily in muscle than in subcutaneous fat. Therefore, diet composition, particularly the protein-to-energy ratio, can be used to increase fatness, which consequently influences performance [50]. Fatteners fed in extensively fed conditions had less proteins compared to intensively fed pigs (150 g/kg vs. 170 g/kg, respectively), whereas the metabolic level was similar (13 MJ ME). Accordingly, our results agree with Lebret et al. [51], who demonstrated that feeding pigs ad libitum with protein- or lysine-deficient but adequate-energy diets during the growing or finishing phases increases IMF proportion.

### 4.3. Fattening System vs. AI, TI, PI, and Fatty acid Profile of Meat

On the basis of the indicators used for estimating a product’s health benefits for humans, we observed a lower atherogenicity index (AI) by 8.2% in the meat fat from Złotnicka Spotted pigs fed extensively with acorn. Although we reported a significantly lower content of C14:0 in meat frompigs fed with acorn, we did not find any differences in AI between both groups to differ statistically (*p* = 0.085). This may be explained by the relationship between the sum of the main saturated fatty acids (including C14:0), which are considered proatherogenic (favoring the adhesion of lipids to cells of the immunological and circulatory system), and the main classes of unsaturated fatty acids, which are considered antiatherogenic, for which we also found statistical differences (n-3 PUFAs and MUFAs) [52]. Another beneficial effect of extensive feeding is the significantly lower (by 12.7%, *p* ≤ 0.05) thrombogenicity index (TI) of the meat from this group of pigs, which shows atendency to form clots in the blood vessels, as a function of the relationship between prothrombogenetic (saturated) and the antithrombogenetic fatty acids [52].

PUFAs(n-3) play a major role in regulating the thrombogenic index, while n-6 PUFAscontribute to the atherogenic index. Myristic (C14:0) and palmitic acids (C16:0) are among the most atherogenic agents, whereas stearic acid (C18:0) is thought to be neutral with respect to atherogenicity but is instead considered to be thrombogenic [53]. In our study, the fat from the *m. longissimus dorsi* of the extensively fed pigs had a significantly higher content of MUFAs, particularly C18:1 n-7, and a lower content of n-3 PUFAs. It was also established that the meat fromthe extensively fed pigs contained over 17% less C14:0 acid (*p* ≤ 0.05), while the amount of C18:0 was 6.7% lower, although, in this case, the difference was not significant. In anexperiment by Rey et al. [6], the meat from Iberian pigs fed with acorns had a high proportion of MUFA C18:1 n-9 (>53–54%) and very low proportions of palmitic (C16:0) and stearic (C18:0) acids (<21% and 9.5%, respectively). Moreover,Cava et al. [3] observed, for the intramuscular total lipids, a significant increase in MUFA content (particularly C18:1), a decrease in SFA content, and no difference in the content of PUFAs in the meat from *montanera* Iberian pigs when compared to Cebo Iberian pigs. Different results were obtained by Hoffman et al. [2] who compared the effect of outdoor and indoor fattening of Landrace × Large White pigs fed a commercial diet without acorns and green forage. The authors demonstrated a significant increase in PUFAs, a decrease in C18:0, and a lack of significant differences in the amount of MUFAs in the meat frompigs kept outdoors. The above results could be an indication that supplementing acorns to the compound feed during the final months of fattening in the *montanera* system may have the greatesteffect on increasing MUFAs in pig meat. Jiménez-Colmenero et al. [54] presented the findings of different authors who observed increases in the content of fat and MUFAs (especially C18:1 acid) in the fat of acorn-fed pigs. Acorns have a high content of the MUFA oleic acid (C18:1 n-9), which accounts for about 60–65% of all acids, whereas grasses, which are usually accessed by free-range pigs, have a predominant content of the PUFA linolenic acid (C18:3 n-3), accounting for around 52–59% of all acids [13]. In view of the above, grass consumed by the pigs may have a decisive impact on increasing the content of n-3 fatty acids in the meat. In our study, the low addition of acorns to the feed did not increase the level of C18:1 (n-9) fattyacid; moreover,the extensively fed pigs had outdoor access, where roughages (grass and whole-crop maize silage) and hay were available. When dried and exposed to air during removal from the silo, silage loses UFAs through oxidation [55]. This may explain the lower level of n-3 PUFAs in the meat from pigs kept in the extensive free-range system in our study, in comparison to free-range fatteners in other studies. Rey et al. [56] showed that feeding pigs with grass in free-range conditions resulted in an increase in not only linolenic acid, but also n-3 C22:5 and C22:6 acids in the intramuscular fat. The same authors also suggested that outdoor rearing allows a higher accumulation of n-3 fatty acids, possibly due to an increased activity of the desaturase and elongase enzymes [56]. Some authors suggested that the concentration of C18:1 n-9, C18:1 n-7, and PUFAs in pig meat is influenced, to a certain degree, by a reduction in dietary protein [43,48]. Wood et al. [48] found that reducing dietary protein content for Large White × Landrace pigs decreased total PUFAs, with an increase in the concentration of MUFAs (C18:1 n-9 and C18:1 n-7) in the *longissimus* muscle. In turn, Tejeda et al. [43] showed that alower protein level in the diet of Iberian pigs caused total n-3 PUFAs to decrease in the *m. serratus ventralis*. However, these authors did not confirm a significant influence of the dietary protein level on the C18:1 n-9 content, ascribing this to the composition of the experimental diets rich in oleic acid. The findings of these authors correspond with the results obtained in our study for C18:1 n-9, C18:1 n-7, and n-3 PUFAs, where the lower dietary protein content in the extensive feeding system likely contributed to the decrease in C18:1 n-7 and n-3 PUFAs in the meat from these fatteners, whereas supplemental acorns (containing 59.36% of C18:1 n-9) caused a slight increase in C18:1 n-9.

The PI index, which indicates susceptibility of meat fat to oxidation, is useful for determining the technological quality of the product. In our study, the PI value was over 25% lower in the meat fromextensively fed pigs, which shows a significantly (*p* ≤ 0.05) lower susceptibility of this meat to rancidity, leading to deterioration of organoleptic traits during storage. This characteristic is particularly valuable for cured meats and traditional long-term matured or dried products, for which the meat from native pig breeds is recommended. The lower susceptibility of the meat to oxidation resulted from the higher amount of MUFAs, which are more resistant to oxidation than PUFAs, especially long-chain ones. Of significance is also the presence of oxidation-reducing tocopherols and polyphenolsin acorns. Whereas this was not evaluatedhere, it is known from other studies that they occur in large amounts in the meat from free-range pigs fed on acorns [7]. These authors observed significantly lower TBARS (Thiobarbituric acid reactive substances) values in the muscles offree-range pigs than in those of pigs raised on concentrate feeds. In addition, the compounds found in acorns have astringent and antidiarrheal properties [57].

## 5. Conclusions

In summary, it may be stated that the meat quality of the native Złotnicka Spotted pig is dependent on the housing and feeding method. Compared to the meat frompigs of the same breed subjected tointensive fattening, the meat fromfree-range pigs extensively fed on silage and small amounts of acorns was characterized by a higher content of fat, which acts as a carrier for flavor and, indirectly, juiciness. The meat from the extensively fed pigs can be described as healthy, due to the significantly higher MUFA content (*p* ≤ 0.05) and the low atherogenic and thrombogenic indices (*p* ≤ 0.05), showing its potential for reducing the risk of atherosclerosis and cardiovascular diseases. Moreover, the significantly (*p* ≤ 0.05) lower peroxidability index (PI) highlights the lower susceptibility of this meat to rancidity and, thus, to deterioration of organoleptic traits during storage, which is particularly valuable for cured meats and traditional long-term matured or dried products, for which the meat from native pig breeds is recommended.

## Figures and Tables

**Table 1 animals-11-00789-t001:** Fatty acid profile (g/100 g of all estimated acids) and lipid quality indices of the feed mixtures and acorns used in different fattening types.

	Conventional Intensive Fattening	Extensive Backyard Fattening	
Item	Feed Mixture	Feed Mixture	Acorns
C6:0	0.52	0.00	0.00
C12:0	0.00	0.00	0.01
C14:0	0.82	0.17	0.16
C15:0	0.00	0.06	0.08
C15:1	0.00	0.00	0.02
C16:0	0.22	12.31	15.87
C16:1 n-9	0.31	0.08	0.00
C16:1 n-7	1.29	0.28	0.08
C17:0	0.25	0.08	0.09
C17:1	0.30	0.00	0.05
C18:0	9.36	2.84	2.05
C18:1 n-9	50.47	29.64	59.36
C18:1 n-7	3.66	1.92	0.65
C18:2 n-6	28.29	43.16	19.42
C18:3 n-3	2.72	8.48	0.82
C20:0	0.61	0.15	0.36
CLA	0.32	0.13	0.00
C20:1 n-9	0.00	0.00	0.32
C20:2 n-9	0.00	0.17	0.05
C20:3 n-6	0.00	0.00	0.10
C20:1	0.00	0.00	0.05
C22:0	0.00	0.00	0.19
C23:0	0.00	0.00	0.08
C24:0	0.00	0.00	0.18
C22:1	0.00	0.34	0.00
C22:5 n-6	0.85	0.00	0.00
Sum of FA	99.99	99.81	99.82
SFA	11.78	15.61	19.07
UFA	88.21	84.20	80.75
MUFA	56.03	32.26	60.53
PUFA	32.18	51.94	20.39
PUFA n-3	2.72	8.48	0.82
PUFA n-6	29.46	43.46	19.52
PUFA n-6/n-3 ratio	10.83	5.13	23.80
PUFA n-3/n-6 ratio	0.09	0.20	0.04
IV	104.18	124.67	-
PI	40.21	61.09	-
MUFA/SFA	4.76	2.07	3.17
PUFA/SFA	2.73	3.33	1.07
UFA/SFA	7.49	5.39	4.23
MUFA/PUFA	1.74	0.62	2.97

SFA—sum of saturated fatty acids: UFA—sum of unsaturated fatty acids; MUFA—sum of monounsaturated fatty acids; PUFA—sum of polyunsaturated fatty acids; IV—iodine value of fat; PI—peroxidability index of fat; 0.00—not detected; sign “-“ denotes that no analysis was performed.

**Table 2 animals-11-00789-t002:** Amino-acid content in the feed mixtures (%/100 g of protein).

	Conventional Intensive Fattening	Extensive Backyard Fattening	
Amino Acids	Feed Mixture	Feed Mixture	Acorns
Threonine	6.38	6.07	2.64
Valine	5.64	6.17	3.56
Methionine	0.71	1.10	0.51
Isoleucine	4.01	4.08	2.85
Leucine	8.61	8.38	5.42
Phenylalanine	6.15	6.42	3.51
Histidine	2.05	1.82	2.46
Lysine	2.74	3.49	4.02
Total EAAs	36.29	37.53	24.97
Aspartic acid	7.30	7.12	9.84
Serine	5.07	4.85	2.95
Glutamic acid	17.03	16.21	10.92
Proline	9.85	9.56	2.95
Glycine	11.62	11.88	2.15
Alanine	4.75	4.75	3.84
Tyrosine	2.59	2.71	1.39
Arginine	5.48	5.41	5.24
Total NEAAs	63.69	62.57	39.28
EAA/NEAA	0.569	0.599	0.636

EAA—essential amino acids; NEAA—nonessential amino acids.

**Table 3 animals-11-00789-t003:** Growing performance and carcass yield of pigs fed according to intensive conventional or extensive traditional systems (means ± SD).

Trait	Conventional Intensive Fattening *n* = 6	Extensive Backyard Fattening *n* = 7
Body weight (kg)	100.83 ^a^ ± 2.79	121.29 ^b^ ± 4.07
Carcass weight (kg)	76.80 ^a^ ± 3.41	95.61 ^b^ ± 4.36
Animal age (m)	9.34 ^a^ ± 0.35	12.00 ^b^ ± 0.00
Cold dressing yield (%)	76.12 ^a^ ± 1.39	78.81 ^b^ ± 1.10
Average daily weight gain (g)	580 ^a^ ± 0.06	340 ^b^ ± 0.01

^a, b^ Mean values in the same row with different letters differ significantly at *p* ≤ 0.05.

**Table 4 animals-11-00789-t004:** Chemical analysis and quality indices of meat (*m.**longissimus dorsi*) (means ± SD).

Item	Intensive Conventional Fattening *n* = 6	Extensive Backyard Fattening *n* = 7
Dry matter (g/kg)	270.5 ± 1.96	283.1 ± 1.13
Water (g/kg)	729.5 ± 1.96	716.9 ± 1.13
Protein (g/kg)	234.3 ± 2.00	232.2 ± 0.59
Fat (g/kg)	21.7 ^a^ ± 0.42	36.0 ^b^ ± 1.14
Ash (g/kg)	11.2 ± 0.02	11.3 ± 0.04
Meat color		
L*	47.86 ± 4.65	49.62 ± 4.94
a*	12.71 ± 1.84	12.14 ± 1.93
b*	12.61 ± 2.02	12.22 ± 2.50

L*—lightness value, a*—redness value, b*—yellowness value. ^a, b^ Mean values in the same row with different letters differ significantly at *p* ≤ 0.05.

**Table 5 animals-11-00789-t005:** Fatty acid composition of *m.longissimusdorsi* (g/100 g of estimated fatty acids) (means ± SD).

Item	Intensive Conventional Fattening (*n* = 6)	Extensive Backyard Fattening (*n* = 7)
C10:0	0.09 ^a^ ± 0.01 ^a^	0.11 ^b^ ± 0.03 ^b^
C12:0	0.13 ± 0.09	0.08 ± 0.01
C14:0	1.58 ^a^ ± 0.23 ^a^	1.30 ^b^ ± 0.07 ^b^
C14:1	0.02 ± 0.01	0.02 ± 0.02
C15:0	0.04 ± 0.03	0.04 ± 0.01
C16:0	27.97 ± 1.74	27.40 ± 1.28
C16:1 n-9	0.64 ^a^ ± 0.06 ^a^	0.72 ^b^ ± 0.09 ^b^
C16:1 n-7	3.40 ^a^ ± 0.16 ^a^	3.34 ^b^ ± 0.65 ^b^
C17:0	0.17 ± 0.04	0.15 ± 0.05
C17:1	0.17 ± 0.03	0.16 ± 0.02
C18:0	13.37 ± 0.29	12.48 ± 1.91
C18:1 n-9	43.55 ± 1.06	44.79 ± 1.96
C18:1 n-7	2.75 ^a^ ± 0.18 ^a^	4.33 ^b^ ± 0.41 ^b^
C18:2 n-6	3.89 ± 0.73	3.44 ± 0.80
C18:3 n-6	0.04 ^a^ ± 0.01 ^a^	0.02 ^b^ ± 0.00 ^b^
C18:3 n-3	0.22 ± 0.05	0.17 ± 0.07
CLA	0.08 ^a^ ± 0.01 ^a^	0.05 ^b^ ± 0.02 ^b^
C20:0	0.15 ± 0.02	0.16 ± 0.02
C20:1 n-9	0.57 ± 0.06	0.62 ± 0.05
C20:2 n-6	0.20 ^a^ ± 0.03 ^a^	0.14 ^b^ ± 0.03 ^b^
C20:3 n-6	0.09 ^a^ ± 0.03 ^a^	0.04 ^b^ ± 0.02 ^b^
C20:4 n-6	0.45 ^a^ ± 0.24 ^a^	0.19 ^b^ ± 0.10 ^b^
C20:4 n-3	0.07 ± 0.01 ^a^	0.03 ^b^ ± 0.02 ^b^
C20:5 n-3	0.01 ± 0.00	0.02 ± 0.01
C22:4 n-6	0.09 ^a^ ± 0.05 ^a^	0.05^b^ ± 0.02 ^b^
C22:5 n-6	0.01 ^a^ ± 0.01 ^a^	0.02 ^b^ ± 0.00 ^b^
C22:5 n-3	0.10 ^a^ ± 0.03 ^a^	0.04 ^b^ ± 0.02 ^b^
C22:6 n-3	0.07 ± 0.03	0.04 ± 0.02
Sum of FA	99.92 ^a^ ± 0.02 ^a^	99.95 ^b^ ± 0.01 ^b^
SFA	43.50 ± 1.60	41.72 ± 3.21
UFA	56.42 ± 1.59	58.23 ± 3.20
MUFA	51.10 ^a^ ± 0.89 ^a^	53.98 ^b^ ± 2.87 ^b^
PUFA	5.32 ± 0.96	4.25 ± 1.00
PUFA n-3	0.47 ^a^ ± 0.10 ^a^	0.30 ^b^ ± 0.11 ^b^
PUFA n-6	4.85 ± 0.92	3.95 ± 0.90
PUFA n-6/n-3 ratio	10.32 ± 2.56	13.17 ± 2.59
PUFA n-3/n-6 ratio	0.10 ^a^ ± 0.02 ^a^	0.08 ^b^ ± 0.01 ^b^
AI	0.61 ± 0.03	0.56 ± 0.05
TI	1.10 ^a^ ± 0.12 ^a^	0.96 ^b^ ± 0.06 ^b^
h/H	1.73 ± 0.14	1.85 ± 0.17
PI	9.77 ^a^ ± 2.07 ^a^	7.25 ^b^ ± 1.64 ^b^
IV	51.54 ± 1.79	53.07 ± 3.08

^a, b^ Mean values in the same row with different letters differ significantly at *p* ≤ 0.05. SFA—sum of saturated fatty acids; UFA—sum of unsaturated fatty acids; MUFA—sum of monounsaturated fatty acids; PUFA—sum of polyunsaturated fatty acids; IV—iodine value of fat; AI—atherogenicity index; TI—thrombogenicity index; PI—peroxidability index of fat; h/H—hypocholesterolemic/hypercholesterolemic ratio.

## Data Availability

All data are available from the authors’ database.

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
