# Peer review of "The Quality and Health-Promoting Value of Meat from Pigs of the Native Breed as the Effect of Extensive Feeding with Acorns"

_animals, 2021, doi:10.3390/ani11030789_

Round 1
Reviewer 1 Report
In this study, Szyndler-Nędza et al aimed to evaluate the meat quality of Złotnicka Spotted pigs from intensive fattening or extensive fattening (with conventional farm-produced compound feed and acorns). The meat of extensively fattened pigs was characterized by a higher content of MUFA and beneficial effect on human health. It’s an interesting study. However, several significant concerns need to be clarified:
General
-Authors need to write the hypnosis in the manuscript.
-The use of letters should be uniform in tables.
-Make sure to define abbreviations correctly; there are many spots throughout the manuscript where a full name is listed but was previously defined.
Specific
L 100-120, In animals, feeding, and housing system, meat quality is a complex trait, and would be affected by slaughter weight, farm conditions, and nutrient levels of diets. What’s the real contribution of acorns to the improvement of meat quality in this study?
L 155, the time of meat color detection should be described in Methods.
L 172, In Statistical analysis, authors need to descript the factors and treatments of the research design.
L 189, Table 1, high levels of PUFA in diets may worsen carcass traits of finishing pigs. The difference in the back fat depth between control and experimental groups should be described.
L 216, Table 3, fat content within pork is significantly influenced by the slaughter weight. Why do intensively or extensively fattened pigs have different slaughter weights? What’s the basis?
Author Response
In the name of all authors I would like to thank reviewer for effort to improve this manuscript. Please find below our comments and answers
General
-Authors need to write the hypnosis in the manuscript.–we added hypothesis
-The use of letters should be uniform in tables. – it was corrected
-Make sure to define abbreviations correctly; there are many spots throughout the manuscript where a full name is listed but was previously defined.– corrected
Specific
L 100-120, In animals, feeding, and housing system, meat quality is a complex trait, and would be affected by slaughter weight, farm conditions, and nutrient levels of diets. What’s the real contribution of acorns to the improvement of meat quality in this study? – in lines 292-294 we wrote “In our study, adding acorns at 5% (per 2 kg of compound feed/head/day) proved inadequate to increase the fat and energy content of the feed, and thus to improve daily weight gains in the extensively fed pigs. Moreover in line 396 we added “low addition of acorns in fed did not influence on increasing level of C18:1 (n-9) fatty acid , moreover”
L 155, the time of meat color detection should be described in Methods. – added
L 172, In Statistical analysis, authors need to descript the factors and treatments of the research design. - added
L 189, Table 1, high levels of PUFA in diets may worsen carcass traits of finishing pigs. The difference in the back fat depth between control and experimental groups should be described – slaughterhouse did not measured this parameter, we only have slaughter weight as well as samples for further analysis however we added „as reported in Table 1 for experimental feeding level of PUFA were higher compared to control. This may cause significant differences in the fatty acid profile of the meat and possible decrease in carcass quality. However, the results of our study indicate that the content of PUFA acids in the meat of extensively fed pigs was not significantly different compared to the control group (Table 5.).
L 216, Table 3, fat content within pork is significantly influenced by the slaughter weight. Why do intensively or extensively fattened pigs have different slaughter weights? What’s the basis? - in intensively feeding pigs faster reach demanded weight 100kg for lean meat. In extensively fattened pigs where finishing weight is 120kg and meat fat is higher are mostly produce for polish regional meat products from pigs.

Reviewer 2 Report
Dear authors, in attached you can find the pdf file with the revision notes.
Best regards.

Author Response
In the name of all authors I would like to thank reviewer for effort to improve this manuscript. Please find below our comments and answers
Line 65-66: the scientific name of the oak must be written in italics every time; there is one parenthesistoo many (Quercuspetraea (Matt.).- it was corrected and names are in italic
Line 73: the first time the meaning of DM must be specified – we added “dry matter (DM)”
Line 91: it is not very correct to write “level of fatty acids in the carcass”; it would be better to rephrasethe sentence. – we rewritten it as “level of meat fatty acids”
Line 106: it is unclear why the control group was fattened for 9 months or up to 100 kg while theexperimental group for 12 months. It would be appropriate to specify this in the text. –we added sentence” Differences between body weight were due purpose of animals – most pigs under preservation programme are kept in extensive condition. For purpose of this research we kept pigs under intensively condition for comparison with extensively feeding regime and with other intensively kept pigs breed”
Line 112: specify the meaning of MJ ME. - megajoules of metabolisable energy
Line 129: Longissimus dorsi must be written in italic and it should be specified between whichvertebrae the cut was made. – corrected and added ”between last thoracic vertebrae and first lumbar vertebrae”
Line 142: write the unit of measurement correctly after 110.- corrected
Line 155-156: Specify the colorimeter model; a space is missing between L * and a *.– added and corrected
Line 189: what is the difference between 0 or 0.00 and/or the “-“ sign? It would be appropriate tostandardize or specify in legend. There is no IV and PI value in the Acorns column, what does thismean?– corrected on 0,00 and added in line 197 “0.00 – not detected, sign “-“no analysis performed.". For acorns we did not count IV and PI, which are indicators estimating a product’s health benefits for humans, because human did not eat raw acorns.
Line 201: it would be better to report the precise value of methionine (0.39 PP) and do not approximate(0.4 PP). – we corrected this as” . Protein from the compound feed fed in the extensive system had a slightly higher content of lysine (3.49 and 2.74, respectively) (by 0.75 percentage points) and methionine (1.10 and 0.71, respectively) – the exogenous amino acids essential for growing pigs”
Line 203: the authors state that: "Protein from the conventional diet was lower in serine", but fromTable 2 it would seem the opposite. – this mistake were corrected – there should be extensively feeding
Line 207: it is not correct to write (% per 100 g of protein), rephrase the sentence. Specify the meaningof EAA / NEAA in legend. – explanation were added in legend and rephrase sentence as “(%/ per 100 g of protein)”
Line 215: too many parenthesis, rewrite like this: (79% vs. 76%; P≤0.05). – corrected as (79% vs. 76%)
Line 216: in Tabs 3, 4 and 5 it must be specified whether it is ± standard error or ± standard deviation;check the spaces between numbers. Superscript the letters of significance. -corrected
Line 224: write Table 4 in bold. - corrected
Line 229: isn't the higher total FA content simply linked to the analytical error, considering that inany case the sum should tend to 100? – yes, analytic error is one of the explanation another is that some fatty acids were on level below detectable level of instrument
Line 229-230: replace the comma with a period (P≤0.05). -corrected
Line 230: the sentence is better understood if “and a statistically…” is replaced with “even if astatistically….”. - corrected
Line 237-241: does it make sense to specify that the atherogenicity index and the C18: 0 are lower
when from a statistical point of view this is not the case? – we removed this part
Line 332-333: correct the term consistent. - corrected
Line 340: couldn't the increase in meat fat simply be due to the age of the animals? – added “also due to animals age”
Line 346-349: the authors state that a diet low in protein or lysine and high in energy increasesintramuscular fat. This statement does not agree with the experimental diet which had 8% less proteinbut also 20% less fat and a higher lysine content. – we added
Fatteners fed in extensively fed had less proteins compared to intensively fed (150 g/kg vs 170g/kg, respectively), and on metabolic level both fed were on same level (13 MJ ME). In this point we agreed with Lebret et al. [51] who demonstrated that feeding pigs ad libitum with protein- or lysine-deficient but adequate-energy diets during the growing or finishing chase increase IMF proportion.
Line 354: this statement does not agree with the statistic (P> 0.05), rephrase the sentence. - corrected

Round 2
Reviewer 1 Report
Animals-1096102-peer-review-v2 is better than animals-1096102-peer-review-v1.
Author Response
Commenst:
Not clear the sentence line 366-372. Please indicate the exact value of P; is it a trend or not significant? If the difference is just numerically lower (that means P > 0.05), sentence must be rewritten. If it is trend change a little and read it carefully
we have corrected sentence to:
Based on the indicators estimating a product’s health benefits for humans, it was observed lower atherogenicity index (AI) by 8.2% in the meat fat of the Złotnicka Spotted pigs fed extensively with acorn. However we reported significant lower content of C14:0 in meat of pigs fed with acorn, we did not find differences for AI between both groups to be differ statistically (P=0.085). This may be explained by relationship between the sum of the main saturated fatty acids (C14:0 is one of them) which are considered pro-atherogenic (favouring the adhesion of lipids to cells of the immunological and circulatory system), and the main classes of unsaturated fatty acids which are considered anti-atherogenic– for which we also found statistical differences (n-3 PUFA and MUFA)